# Transplantation of Mesenchymal Stem Cells Derived from Old Rats Improves Healing and Biomechanical Properties of Vaginal Tissue Following Surgical Incision in Aged Rats

**DOI:** 10.3390/ijms25115714

**Published:** 2024-05-24

**Authors:** Ofra Ben Menachem-Zidon, Benjamin Reubinoff, David Shveiky

**Affiliations:** 1The Sidney and Judy Swartz Stem Cell Research Center, The Goldyne Savad Institute of Gene Therapy, Hadassah Medical Center and Faculty of Medicine, Hebrew University of Jerusalem, Jerusalem 91120, Israel; ofra.benmenahem@mail.huji.ac.il; 2Department of Obstetrics and Gynecology, Hadassah Medical Center and Faculty of Medicine, Hebrew University of Jerusalem, Jerusalem 91120, Israel; rdavis@hadassah.org.il

**Keywords:** aging, mesenchymal stem cell transplantation, pelvic floor dysfunction

## Abstract

Pelvic floor dysfunction encompasses a group of disorders that negatively affect the quality of women’s lives. These include pelvic organ prolapse (POP), urinary incontinence, and sexual dysfunction. The greatest risk factors for prolapse are increased parity and older age, with the largest group requiring surgical intervention being post-menopausal women over 65. Prolapse recurrence rates following surgery were reported to be as high as 30%. This may be attributed to ineffective healing in the elderly. Autologous stem cell transplantation during surgery may improve surgical results. In our previous studies, we showed that the transplantation of bone marrow-derived mesenchymal stem cells (MSCs) from young donor rats improved the healing of full-thickness vaginal surgical incision in the vaginal wall of old rats, demonstrated by both histological and functional analysis. In order to translate these results into the clinical reality of autologous MSC transplantation in elderly women, we sought to study whether stem cells derived from old donor animals would provide the same effect. In this study, we demonstrate that MSC transplantation attenuated the inflammatory response, increased angiogenesis, and exhibited a time-dependent impact on MMP9 localization. Most importantly, transplantation improved the restoration of the biomechanical properties of the vagina, resulting in stronger healed vaginal tissue. These results may pave the way for further translational studies focusing on the potential clinical autologous adjuvant transplantation of MSCs for POP repair for the improvement of surgical outcomes.

## 1. Introduction

Pelvic floor dysfunction (PFD) is a group of disorders that negatively affect the quality of women’s lives. These include pelvic organ prolapse (POP) [1], urinary incontinence [2], and sexual dysfunction [3]. POP represents a major health issue for women worldwide and affects nearly 50% of the population of post-menopausal women [4,5]. Vaginal delivery and age are the main risk factors for POP [6,7,8]. Current surgical treatments for pelvic floor disorders using either the patients’ own tissues or synthetic grafts yield suboptimal anatomical outcomes [9,10,11], with reoperation rates greater than 30% in some studies [12]. Therefore, there is an urgent need for new treatment approaches.

Mesenchymal stem cells (MSCs) have the ability to proliferate to some extent in culture, and to differentiate into various cell types, such as muscle, bone, cartilage, and collagen-producing fibroblasts [13,14,15]. These properties, along with the ability of transplanted MSCs to migrate to sites of injury, to modulate inflammatory responses and other effects are the basis for their potential use in regenerative medicine and tissue engineering [16]. In addition, their autologous transplantation evades immune rejection and the need for immunosuppressive therapy [17]. However, the regenerative therapeutic potential of stem cells harvested from elderly people is controversial [18,19,20,21,22].

To mimic vaginal healing post pelvic floor surgery, we established a novel rodent model of vaginal surgical injury that enables the study of vaginal healing and the potential adverse effects of aging [23,24]. Using our model, we demonstrated the impaired healing of the surgical injury and weakened biomechanical properties of the healed vaginal tissue in aged rats. The systemic transplantation of bone marrow (BM)-derived MSCs after vaginal incision was associated with homing and survival of the MSCs at the injury site for 30 days. MSC transplantation attenuated the inflammatory response, increased angiogenesis, and attenuated MMP9 immunolocalization at the injury site. Most importantly, transplantation improved the restoration of the biomechanical properties of the vagina, resulting in stronger healed vaginal tissue [25]. In our pursuit of translating these results into the clinical application of autologous MSC transplantation in elderly women undergoing POP repair surgery, the aim of this study was to examine whether MSCs derived from old donor rats can yield equivalent therapeutic effects on vaginal wound healing following surgical incision in inbred aged rats.

## 2. Results

### 2.1. Homing and Survival of Transplanted Old-Donor-Derived MSCs Following Full-Thickness Vaginal Incision

Old Sprague Dawley (SD) rats underwent a full-thickness vaginal incision. The incision included both the vaginal epithelial layer and the fibromuscular tissue underneath. Healing was analyzed and compared between sham-operated rats and MSC-transplanted rats on day 3 post injury. The incision was almost fully bridged with granulation tissue and epithelium in the old MSC-transplanted group (Figure 1B), as opposed to initial healing in sham-operated old control rats (Figure 1A). The evaluation of the maximal distance between the epithelial edges of the wound on day 3 showed a significantly smaller gap (*p* < 0.01) in the old MSC-transplanted rats compared with the sham non-transplanted controls (Figure 1C). PKH-26-labeled MSCs homed to the vaginal injury site and were present in the lamina propria at day three post transplantation (Figure 1D–F). These results are in accordance with our previous publications, in which we demonstrated the homing of systemically transplanted MSCs to vaginal injury sites. Most importantly, when MSCs were systemically transplanted to naïve rats with no injury, no MSCs were found in the vaginal tissue [1,2].

### 2.2. The Effect of Old-Donor-Derived MSC Transplantation on the Inflammatory Response at the Vaginal Surgical Site

We studied the potential of old-donor-derived MSC transplantation to modulate the inflammatory response at the vaginal surgical injury site. We assessed inflammation by immunofluorescence staining for CD68, a pan-macrophage marker. Sections were stained and analyzed for macrophages expressing CD68 around the injury site. The quantification of the macrophage numbers in the sham-operated rats and MSC-transplanted rats on day 3 post injury showed a significantly higher number of CD68+ cells in the sham-operated rats compared with the old MSC-transplanted rats. This difference was also seen on day 30 post transplantation (Figure 2A). To further characterize the inflammatory response, sections were co-stained for CD68 and tumor necrosis factor α (TNF-α). As early as day 3, the percentage of the total CD68+ cells co-expressing TNF-α was significantly lower in the old MSC-transplanted rats compared with the sham-operated rats. This difference was also significant on day 30 post injury (Figure 2B). Importantly, we did not observe any long-term local side effects throughout the 30-day follow-up of the transplanted rats.

Taken together, these results showed that old-donor-derived MSC transplantation was associated with the attenuation of the inflammatory response at days 3 and 30 post injury.

### 2.3. The Effect of Old-Donor-Derived MSC Transplantation on Acute and Chronic MMP9 Expression Following Injury

MMP9 is a key enzyme involved in the degradation of both collagen and elastin in the extracellular matrix (ECM). An analysis of the expression of MMP9 in the vagina was performed by immunofluorescence staining. On day 3 post transplantation, the sham-operated rats displayed strong and scattered expression of MMP9 (Figure 3A), while the MSC-transplanted old rats displayed decreased expression of MMP9 (Figure 3B). The quantification of MMP9+ cells on day 3 post injury revealed significantly higher numbers of MMP9+ cells in the sham rats compared with the MSC-transplanted rats (Figure 3D). We next examined the expression of MMP9 on day 30 post transplantation. On day 30, MMP9 expression was almost absent in the sham-operated rats but was still present in the MSC-transplanted rats (Figure 3C,D).

### 2.4. The Effect of MSC Transplantation on Blood Vessel Number on Day 30 Post Surgery

In our previous studies in young rats [1] and in old rats transplanted with young-donor-derived MSCs [2], MSC transplantation was associated with increased blood vessel formation at the vaginal injury site. The increased blood vessel formation occurred either through a trophic effect on angiogenesis in the recipient or the differentiation of the transplanted MSCs, contributing to the formation of either postcapillary venules, precapillary arterioles, or venules and arterioles. We therefore quantified the number of blood vessels at day 30 post transplantation, looking specifically for blood vessels containing PKH-26 cells within the blood vessel wall. MSC-transplanted rats displayed significantly more blood vessels, compared to both naïve and sham-operated rats (Figure 4A). However, in an analysis of five random sections from seven rats, we did not observe PKH-26-expressing cells within the walls of the blood vessels, but adjacent to them, distributed at the lamina propria (Figure 4B–E).

### 2.5. The Effect of MSC Transplantation on Biomechanical Properties of the Vaginal Tissue

On day 30 post transplantation, the vagina was dissected and prepared for biomechanical evaluation. The healed vagina in its full thickness was loaded onto a mechanical analyzer, and tension was gradually increased until the breaking point was reached. The MSC-transplanted rats displayed significantly higher peak forces (N = 4 ± 0.2) than the sham-operated rats (N = 2.9 ± 0.3; *p* < 0.05, Figure 5A). The beneficial effect of MSC transplantation was also observed in the stress–strain curve (Figure 5B). The vaginal tissues from the sham-operated rats displayed a shorter stretching time (18.8 ± 3.6), while the MSC-treated rats displayed a significantly longer stretching time (43.1 ± 3.9; *p* < 0.05; Figure 5C). Looking at the length of time at a high load, the vaginal tissue from the sham-operated rats was stretched for a significantly shorter time at a high load (17 ± 4) compared with the MSC-treated rats (37 ± 6; *p* < 0.01) (Figure 5D).

## 3. Discussion

In this study, we demonstrated that the systemic transplantation of old-donor-derived MSCs after vaginal full-thickness surgical incision in aged rats was associated with the homing and survival of the MSCs at the injury site for 30 days. The MSC transplantation attenuated the inflammatory response, increased angiogenesis, and affected MMP9 immunolocalization at the injury site in a time-dependent manner. Most importantly, transplantation improved the restoration of the biomechanical properties of the vagina, resulting in stronger healed vaginal tissue.

In our previous studies, we demonstrated that the systemic transplantation of young-donor-derived bone marrow MSCs after vaginal full-thickness incision in old rats was effective [2]. Given that pelvic organ prolapse predominantly affects older women, the primary objective of this study was to assess the specific efficacy of stem cells derived from old donors, aiming to provide insights into their suitability for therapeutic applications in pelvic organ prolapse.

Previous studies in different animal models of regeneration suggested that the regenerative therapeutic effect of MSCs occurs early on in the days following transplantation. Our findings corroborated the above, since enhanced wound closure kinetics were evident as early as three days after MSC transplantation.

The effect of transplanted MSCs might occur by different mechanisms and during different wound healing phases: (i) Immunomodulation [3]. (ii) The secretion of paracrine factors promoting the healing processes of fibroblasts and keratinocytes, the modulation of the extracellular matrix, and blood vessel formation. (iii) MSC differentiation into tissue-specific cells [3,4]. In addition, it is not clear how the age of the donor of the MSCs affects each of these mechanisms [5,6,7,8]. In the present study, we evaluate, for the first time, the regenerative potential of the transplantation of aged-donor-derived MSCs in vaginal wound healing following full-thickness surgical incision in aged rats.

MSCs have been shown to exhibit immunomodulatory properties at the sites of inflammation. An age-related decline in the expression of inflammatory response genes may indicate that MSCs undergo an age-related decline in their immunomodulatory activity [10]. However, Siegel et al. [11] found that biological age did not affect the ability of human MSCs to suppress the proliferation of activated allogeneic T-cells in vitro. Similarly, in our model, old-donor-derived MSC transplantation attenuated the macrophage number as well as the percentage of TNFa-expressing cells. Since the immunomodulatory effect of MSCs may depend on multiple molecular pathways and mechanisms, it has been suggested that some are affected by age, while others are not [8,11,12,13]. While our data support the maintenance of the immunomodulatory potential of MSCs at an advanced age, further studies are needed to dissect the effect of age on the various molecular mechanisms that play a role in the immunomodulatory properties of MSCs.

Angiogenesis is an important process in wound healing, with the generation of blood vessel-rich granulation tissue being a critical step in tissue regeneration. The transplantation of bone marrow (BM)-derived MSCs has been suggested as a potential clinical approach to promote therapeutic angiogenesis. Several studies have suggested that old-donor-derived BM MSCs have reduced angiogenic potential [15], while other studies have claimed that the induction of angiogenesis by MSCs is not dependent on the donor age [16]. Our data supported the view that the angiogenic effect of the MSCs was maintained with age. The increase in the number of blood vessels at the healed vaginal site was similar to the increase observed in our previous study of the transplantation of young donor-derived BM MSCs in aged rats [2]. Andrzejewska et al. (2019) [15] suggested that in vitro aging rather than in vivo aging exerted a strong influence on the cellular properties of BM-derived MSCs, including the pro-angiogenic effect. Since sufficient BM MSC numbers can only be obtained by extensive expansion, this might be the more significant limiting factor for using BM MSCs in cellular therapy. In our study, the BM MSCs were not cultured for prolonged periods, and further studies are needed to determine whether more extensive expansion would hamper their angiogenic effect.

The effect of donor age on the differentiation potential of BM MSCs is controversial, and while several studies claim that donor age has no effect [5,10], others report an age-related shift in the differentiation from an osteogenic to adipogenic fate [16,17,18,19]. This shift may be explained by the significant decline in sex hormone levels, which occurs in aging and is specifically relevant to women [7,10]. Although growing evidence suggests that BM-derived MSCs have the ability to differentiate into endothelial cells [20,21,22,23], the effect of donor age on the potential of MSCs to differentiate into endothelial cells was not studied. In our previous study, transplanted young-donor-derived BM MSCs differentiated to CD31+ endothelial cells within the walls of newly formed blood vessels at the healing vaginal injury site of old rats. Here, we could not demonstrate the endothelial differentiation of old-donor-derived MSCs in vivo. It would be interesting, in future studies, to determine whether the endothelial differentiation of BM MSCs is also hampered by the donor age in vitro. In our model, the change in the differentiation potential did not seem to affect the functional therapeutic outcome of old-donor-derived MSCs.

Several studies on human pathological samples have shown that POP is associated with an increase in the expression of MMPs, including MMP1, MMP2, and MMP9, and decreases in the activity of TIMP1-4, leading to tissue degradation. In addition, it has been shown that MMP-9 is upregulated immediately after wounding and appears to delay wound healing through interference with re-epithelialization [25]. In accordance with these observations, we demonstrated the induction of MMP9 expression on day 3 following vaginal injury. It may be hypothesized that the observed MSC transplantation-induced reduction in MMP9 expression on day 3 may be associated with the decreased degradation of vaginal ECM components, resulting in the improved biomechanical properties of the treated vagina. On day 30, the sham-operated rats displayed low levels of MMP9 compared with the MSC-transplanted rats. It is possible that although higher levels of MMP-9 at the acute phase are harmful, the elevation of its expression at later stages may be beneficial. This is in accordance with a previous study claiming that reduced levels of MMPs at different phases of wound healing are associated with delayed epithelialization [25,26]. Taken together, our results are in agreement with other studies [27,28,29], suggesting that MMP-9 is vital for successful wound healing, coordinating and affecting multiple events involved in the process of epithelial regeneration.

A key finding of our study was that old-donor-derived MSCs improved the biomechanical properties of the healed vagina, including significantly increased maximal tensile strength 30 days post injury and transplantation. It may be hypothesized that the observed MSC-induced effect on inflammation, MMP9 expression, and angiogenesis all contribute to the functional biomechanical effect. 

In conclusion, we showed that the systemic transplantation of BM-derived MSCs, isolated from old rats, improved the healing of vaginal tissue following full-thickness surgical incision. These results may pave the way for the potential clinical application of autologous MSC transplantation in elderly patients undergoing pelvic reconstructive surgery. The local transplantation of MSCs may provide clinical advantages compared with systemic delivery, as it avoids their systemic dissemination and potential undesired side effects [30]. However, our previous studies [31], and those of others [32,33,34,35], showed that the local injection of MSCs into the vagina resulted in the poor survival of the transplanted cells. This may be due to various factors, including the host environment and, importantly, the method of local delivery, which is currently not optimal. We therefore transplanted the MSCs systemically and showed their homing and long-term survival in the injured vagina. Future developments in the delivery systems of MSCs may allow for simple and safe local transplantation combined with cell survival and the beneficial healing effects of systemic transplantation.

## 4. Materials and Methods

This study was approved by the Hebrew University Animal Care and Use Committee. SD female rats were held in the specific pathogen-free unit in Hadassah Hebrew University Medical School with food and water ad libitum. Old rats were 14–16 months old with an average weight of 600–700 g. The rats were divided into the following three groups: naive rats, which did not undergo any surgical procedure or treatment; sham rats, which underwent a full-thickness vaginal incision and received the medium systemically, immediately following the vaginal incision; and MSC-transplanted rats, which received systemic i.v. MSC transplantation, following full-thickness vaginal incision.

Isolation of old bone marrow-derived MSCs and labeling the cells before transplantation:

MSCs were isolated according to [36]. Briefly, the tibia and femur bones of 14-month-old female SD rats were isolated, and the bone marrow was flushed with 25 mL DMEM. The cells were centrifuged at 1200 RPM for 10 min at 4 °C and resuspended with 20 mL DMEM containing 10% FBS, 1% pen-strep, and 1% glutamine (MSC medium, all products from Gibco, Modi’in, Israel). A total of 10 mL of cell suspension was seeded into a 10 cm culture Petri dish and further incubated in a 5% CO_2_, 37 °C incubator. Medium was changed 24 h afterward and, from then on, every 2–3 days. PKH-26 labeling was performed at passage 4 prior to transplantation. Cells were dissociated by trypsin EDTA and then incubated with PKH-26 according to the recommended protocol by Sigma (PKH26GL, Sigma, Rehovot, Israel). Successful labeling and the percentage of labeled cells were verified before each transplantation using fluorescence microscopy and flow cytometry.

### 4.1. Animal Model and Procedure

Following administration of anesthesia using a mixture of ketamine and domitor (75 mg/kg BW [body weight] and 0.5 mg/kg BW), a standardized 9 mm posterior midline sagittal full-thickness vaginal incision was performed opposite to the urethra. A total of 2 × 10^6^ MSCs were systemically transplanted into the tail vein immediately following the full-thickness vaginal incision. Upon completion of the procedure, administration of atipamezole (1 mg/kg) was given to reverse the anesthetic effects. Rats were monitored daily after the incision. According to the protocol, in case of excessive bleeding, rats were excluded from the experiment. For histological analysis, including inflammatory response, rats were sacrificed at 3, 7, and 30 days post transplantation (n = 7–9 rats/group/at each time point). We quantified the number of labeled MSCs in the injured vagina at days 3, 7, and 30 after transplantation. We examined every fifth sequential slide covering the tissue surrounding the entire vaginal injury. Cell counts from serial sections were corrected according to the Abercrombie method [37] and adjusted to the total size of the graft. It should be noted that the labeled MSCs were observed near the injury site while more peripheral tissues in the scored slides did not include labeled cells. Hence, the quantitative analysis included the entire distribution of labeled MSCs in the tissues surrounding the injury site.

For biomechanical studies, rats were sacrificed at 30 days post transplantation.

### 4.2. Dissection of Vaginal Tissue and Histological Evaluation

Rats were sacrificed with an overdose of CO_2_. In sacrificed rats, the entire vagina including the adjacent extra vesicle urethra was dissected en bloc and carefully embedded in paraffin, preserving its anatomical orientation. Six-micrometer sections were prepared and used for all histological parameters. H&E staining was performed on every ninth section to identify and map the grafts’ location and for evaluation of wound diameters. Quantification of PKH26-positive cells in the injured vagina at days 3, 7, and 30 after transplantation was performed on every fifth sequential slide covering the tissue surrounding the entire vaginal injury. Cell counts from serial sections were corrected according to the Abercrombie method and adjusted to the total size of the graft. For inflammation assessment, five sections from each rat (n = 7 rats/group) were co-stained with anti-CD68 and anti-TNF-α. CD68^+^ cells and those co-expressing TNF-α were quantified in three random fields of vaginal tissue adjacent to the vaginal incision at ×20 magnification. The analyzed fields did not include the lumen. For vessel formation assessment, sections were stained with either anti-CD31 or anti-SMA. Quantification of the total number of blood vessels with a diameter above 10 μm in the lamina propria surrounding the injury site was quantified on days 3, 7, and 30 in old rats. For MMP9 expression, immunofluorescence staining was performed with anti-MMP9. MMP9-expressing cells were observed and quantified at ×20 magnification in the lamina propria adjacent and surrounding the epithelial layer of the injury site. Quantification of blood vessels per field and of MMP9-expressing cells was performed in three random fields in five sections from each rat (n = 5 rats/group) as above. Incubation with all primary antibodies was for 24 h at 4 °C, followed by three washes with 1 × PBS (Beit Haemek, Israel), and then the appropriate secondary antibodies (all from Jackson Laboratories, Petach Tikvah, Israel) were added (see details of antibodies in Appendix A).

### 4.3. Biomechanical Testing

Biomechanical properties of the vaginal tissue were tested using a dynamic mechanical analyzer (TA Instruments, New Castle, DE, USA). The load cell used was of 22 N with a resolution of ±0.1 N. Thirty days after injury, rats (n = 9 in each group) were euthanized and the vaginal specimen was longitudinally cut at the midline of the anterior wall along the urethra, forming a rectangular shape with the healed wound in the middle. A standardized rectangular specimen sized 1 × 2 cm was prepared for biomechanical testing. The specimens were kept moist by spraying sterile warm (37 °C) saline solution prior to and during testing. Each specimen was loaded onto the mechanical analyzer using a smooth plastic clench, and tension was gradually increased until the breaking point was reached. The induced force was tracked at a 5% strain rate up to the breaking point of the subject tissue, and we observed changes in the stress–strain curve. The following parameters were analyzed: (1) maximal tensile strength, represented by the peak force (N) applied to each sample just before tearing. (2) A stress–strain curve was drawn for each specimen to measure elastic properties. Stress (Pa) was defined as the force applied to the tissue divided by the cross-sectional area of the specimen. Strain was defined as the percentage of change in length of the specimen compared with the initial length between the clamps. (3) Viscoelastic properties are presented as the time (sec) elapsed at maximal load under a continuingly increasing strain (time in high load). This is a dynamic measure of tissue strength, looking at the ability of the tissue to maintain its biomechanical properties under strain over time.

### 4.4. Statistical Analysis

Normality and equal variances tests were performed to ensure the appropriate use of parametric tests. Data are presented as mean ± SD. One-way ANOVA was performed on these data. Post hoc analysis between specific groups was performed using Bonferroni correction. *p* values of <0.05 were considered significant.

## Figures and Tables

**Figure 1 ijms-25-05714-f001:**
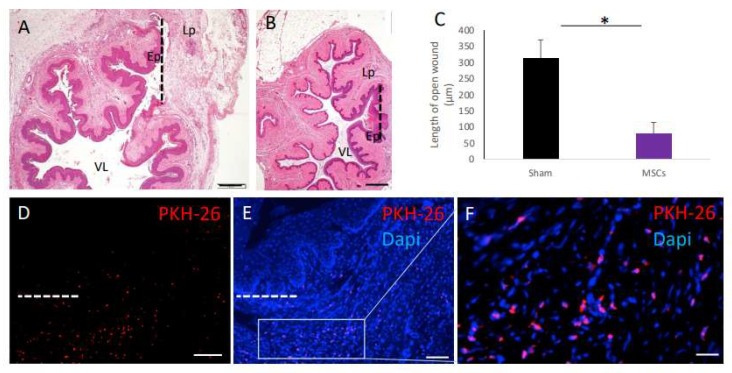
Homing and survival of transplanted MSCs in the vaginal surgical site. (**A**,**B**) Low-power images of H&E staining show the location of the injury site in sham- (**A**) and MSC-transplanted (**B**) old rats at day 3 post transplantation. The dashed line marks the location of the vaginal surgical incision site. (**C**) Bar graphs displaying the maximal distance between epithelial edges of the wound (μm) in sham- and MSC-transplanted rats at 3 days post transplantation (*p* < 0.05). Transplanted PKH-26-labeled MSCs at the lamina propria (**D**) and DAPI-stained fluorescence image (**E**) of the same section. A higher magnification of PKH-26-labeled MSCs at the lamina propria is shown in (**F**). The dashed line marks the location of the vaginal surgical incision site. VL, vaginal lumen; Lp, lamina propria; Ep, epithelium. Scale bars: (**A**,**B**) = 500 µm; (**D**,**E**) = 200 µm; (**F**) = 50 µm. * *p* < 0.05.

**Figure 2 ijms-25-05714-f002:**
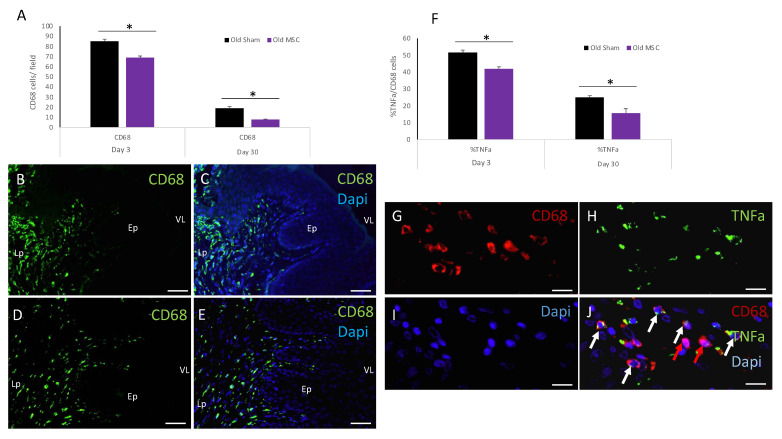
The effect of MSC transplantation on the inflammatory response at the vaginal surgical site. (**A**) Quantification of CD68+ cells in sham- and MSC-transplanted rats at 3 and 30 days post injury and transplantation (*p* < 0.05). Representative immunofluorescence images of macrophages expressing CD68 (green) around the injury site from a sham-operated rat (**B**) and an MSC-transplanted rat (**D**) at day 3 post injury. Nuclei are counterstained with DAPI ((**C**,**E**), respectively). (**F**) Bar graphs displaying the percentage of CD68+ cells co-expressing TNF-α from total CD68+ cells in sham- and MSC-transplanted rats at 3 and 30 days post transplantation (*p* < 0.05). Images of CD68 cells ((**G**); red) and TNF-α ((**H**); green) around the injury site, counterstained with DAPI (**I**). (**J**) The merged images were used to quantify the percentage of the CD68-expressing cells that were TNFα+ cells. Cells co-expressing CD68 and TNF-α are marked by white arrows. The red arrow shows CD68+ cells that do not express TNF-α. VL, vaginal lumen; Lp, lamina propria; Ep, epithelium. Scale bars: (**B**–**E**) = 200 µm; (**G**–**J**) = 50 µm. * *p* < 0.05.

**Figure 3 ijms-25-05714-f003:**
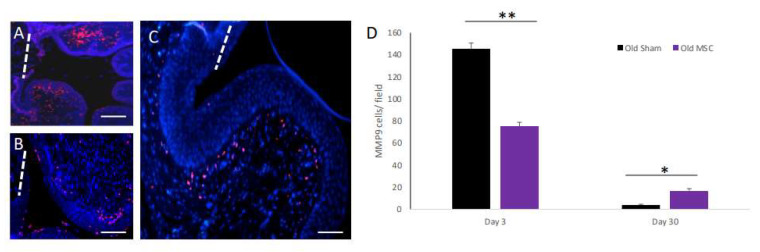
The effect of MSC transplantation on MMP9 expression following vaginal surgical incision. Representative images of sections from sham- (**A**) and MSC-transplanted (**B**) old rats at day 3 post transplantation stained with anti-MMP-9 antibodies (red), and counterstained with DAPI (blue). The dashed white line marks the location of the vaginal injury site. (**C**) Representative image of a section from an MSC-transplanted old rat at day 30 post transplantation immuno-stained for MMP-9. (**D**) Bar graphs displaying quantification of MMP9+ cells on days 3 and 30 post injury. Scale bars: (**A**,**B**) = 500 µm; (**C**) = 100 µm. (* *p* < 0.05; ** *p* < 0.01).

**Figure 4 ijms-25-05714-f004:**
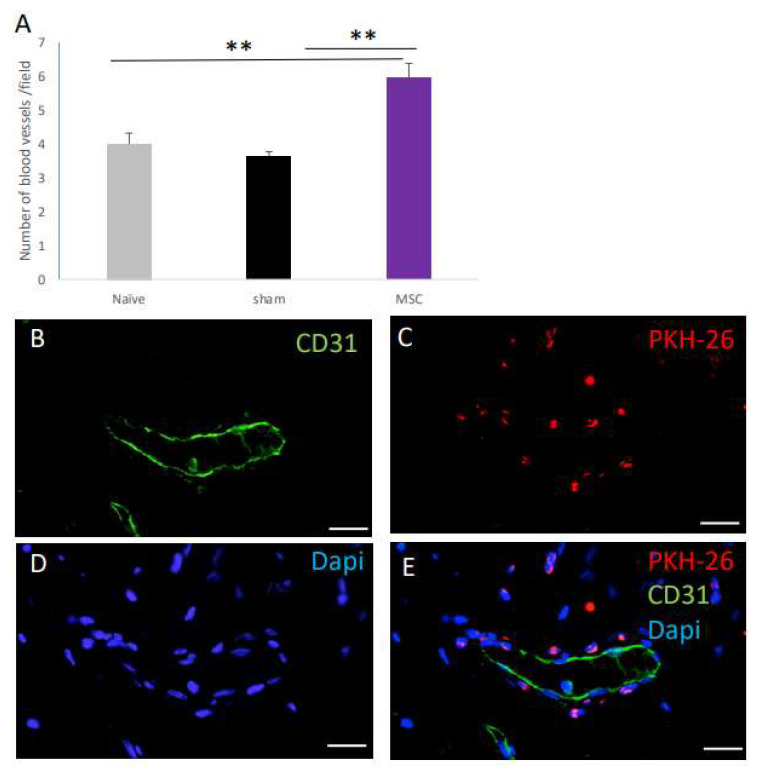
The effect of MSCs on blood vessel number on day 30 post-surgery. (**A**) Bar graphs displaying the total number of blood vessels (including postcapillary venules, precapillary arterioles, venules, and arterioles) per microscopic field in old naïve and sham- and MSC-transplanted rats at 30 days post transplantation (*p* < 0.0501). Representative images of a postcapillary venule/precapillary arteriole that is CD31+ (**B**), surrounded by PKH-26+ cells (**C**) and counterstained with DAPI (**D**). A merged image is presented in (**E**). Scale bars: (**B**–**E**) = 50 µm. ** *p* < 0.01.

**Figure 5 ijms-25-05714-f005:**
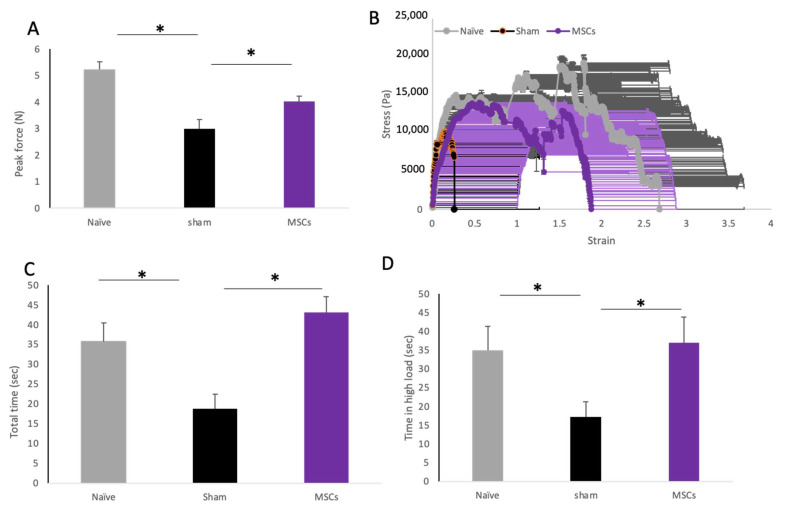
The effect of MSC transplantation on biomechanical properties of the vaginal tissue. (**A**) Bar graphs displaying the maximal tensile strength (N) of vaginal tissue by study group. (**B**) Stress–strain curves of old naïve and sham- and MSC-transplanted rats. Mean stress (Pa) values ± SD are presented for each experimental group. (**C**) Bar graphs displaying the time (seconds) of stretching before tear of vaginal tissue from each group. (**D**) Bar graphs displaying the length of time (seconds) at peak stress, until loss of tensile strength by study group, representing viscoelasticity. * *p* < 0.05.

## Data Availability

The data that support the findings of this study are available from the corresponding author upon request.

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
