# Peer review of "Transplantation of Mesenchymal Stem Cells Derived from Old Rats Improves Healing and Biomechanical Properties of Vaginal Tissue Following Surgical Incision in Aged Rats"

_ijms, 2024, doi:10.3390/ijms25115714_

Round 1

Reviewer 1 Report

Comments and Suggestions for Authors

How do authors enusre the homing of systemically adminstered MSC's to the vaginal site?

What was the number of cells administered? 

Whether only single dose was given? How the prolonged effect took place with a single systemic dose of cells?

Reviewer 2 Report

Comments and Suggestions for Authors

Pelvic floor dysfunction can negatively affect women quality of life and it is a relative frequent problem in women over 65 years.

The authors developed a novel rodent model of vaginal surgical injury to investigate vaginal healing and the potential negative effect of aging. They previously reported that systemic transplantation of mesenchymal stem cells (MSCs) from young donors improved healing of vaginal incision, and have a beneficial effect on functionality.  Aim of this study was to demonstrate analogous effect after auto transplantation of MSCs from old donor rats, so supporting its translation into clinics and they substantially confirmed that an improvement of healing and of vaginal function can be also be obtained with MSCs from old donor. 

The paper is very interesting, experiments well projected and described.

I have just few questions:

·      How did they establish  the number of MScs? In rat experiments the used a single fix dose. Should be so also in the clinic? 

·      How did they explain the reduced survival of MSCs after direct intravaginal injection?

·      Did they observed long term local side effects due to MSCs, such as calcifications tissue degeneration ?
